

# Is perfectionism associated with academic burnout through repetitive negative thinking?

David Garratt-Reed, Joel Howell, Lana Hayes and Mark Boyes

School of Psychology, Curtin University, Australia

## ABSTRACT

Academic burnout is prevalent among university students, although understanding of what predicts burnout is limited. This study aimed to test the direct and indirect relationship between two dimensions of perfectionism (Perfectionistic Concerns and Perfectionistic Strivings) and the three elements of Academic Burnout (Exhaustion, Inadequacy, and Cynicism) through Repetitive Negative Thinking. In a cross-sectional survey, undergraduate students ($n = 126$, $M_{age} = 23.64$, 79% female) completed well-validated measures of Perfectionism, Repetitive Negative Thinking, and Academic Burnout. Perfectionistic Concerns was directly associated with all elements of burnout, as well as indirectly associated with Exhaustion and Cynicism via Repetitive Negative Thinking. Perfectionistic Strivings was directly associated with less Inadequacy and Cynicism; however, there were no indirect associations between Perfectionistic Strivings and Academic Burnout operating through Repetitive Negative Thinking. Repetitive Negative Thinking was also directly related to more burnout Exhaustion and Inadequacy, but not Cynicism. It is concluded that future research should investigate whether interventions targeting Perfectionistic Concerns and Repetitive Negative Thinking can reduce Academic Burnout in university students.

## INTRODUCTION

Individuals experiencing burnout have difficulty committing to the task at hand, feel detached and dissatisfied with their work, and are less productive (*Leiter & Maslach, 2003*). Research has identified three main dimensions of burnout: Exhaustion, Cynicism, and Inadequacy. Exhaustion refers to the stress that leaves an individual feeling unable to commit to the task at hand (*Maslach & Jackson, 1981*). Cynicism refers to an individual's cynical attitude toward work, which leads to negative, detached feelings towards their work (*Leiter & Maslach, 2003*). Inadequacy refers to the individual feeling incompetent at work, which is often accompanied by reduced productivity and dissatisfaction with their work achievements (*Leiter & Maslach, 2003*). Whilst much research has studied burnout in the workplace, recent research is exploring how these aspects of burnout present in university students, which is important given evidence that Academic Burnout is associated with poor educational outcomes (*Mazurklewicz et al., 2011*; *Zhang, Gan & Cham, 2007*).

Corresponding author
Joel Howell,
joel.howell@curtin.edu.au

Similar to burnout in other contexts, Academic Burnout involves three elements: *Exhaustion* due to study demands; feelings of *Inadequacy* as a student due to the long-term stress of striving for academic achievement; and a *Cynical Attitude* toward study (*Merino-Tejedor et al., 2014*; *Salmela-Aro et al., 2009*). Academic burnout is associated with various negative consequences, including poor academic outcomes, increased psychological distress, reduced life satisfaction, and sleep deprivation (e.g., *May, Bauer & Fincham, 2015*; *Mazurklewicz et al., 2011*; *Salmela-Aro & Upadyaya, 2014*). Given that Academic Burnout is highly prevalent among university students (e.g., *Kristanto, Chen & Thoo, 2016*; *Mazurklewicz et al., 2011*), it is important to develop methods of reducing Academic Burnout. A vital step towards this is to identify factors associated with burnout. Perfectionism has been linked to numerous psychological symptoms (e.g., *Limburg et al., 2017*) and preliminary evidence also indicates that it predicts Academic Burnout (*Kljajic, Gaudreau & Franche, 2017*). Additionally, there is emerging evidence that the relationship between perfectionism and various psychological symptoms is mediated by Repetitive Negative Thinking (*Egan, Hattaway & Kane, 2014*). Consequently, the current study will investigate whether two forms of perfectionism (Perfectionistic Concerns and Perfectionistic Strivings) predict the elements of Academic Burnout, and whether these relationships are mediated by Repetitive Negative Thinking.

Perfectionism has been implicated in the development and maintenance of a variety of psychopathologies (e.g., *Limburg et al., 2017*) and is also associated with Academic Burnout (*Kljajic, Gaudreau & Franche, 2017*). Across multiple perfectionism scales, there are two higher order dimensions: Perfectionistic Concerns and Perfectionistic Strivings (*Burgess, Frost & DiBartolo, 2016*; *Limburg et al., 2017*; *Stoeber & Otto, 2006*). Perfectionistic Concerns involves being overly concerned about mistakes in performance as well as doubting one's actions, and is consistently related to negative psychological outcomes (*Bieling, Israeli & Antony, 2004*; *Limburg et al., 2017*). Across two of the most popular perfectionism scales the Concern over Mistakes, Doubts about Actions, Parental Expectations, and Parental Criticism Subscales from the Frost Multidimensional Perfectionism Scale (FMPS; *Frost et al., 1990*), and Socially Prescribed Perfectionism from the Hewitt and Flett Multidimensional Perfectionism Scale (HMPS; *Hewitt & Flett, 1991*) consistently load onto the Perfectionistic Concerns factor (*Bieling, Israeli & Antony, 2004*; *Limburg et al., 2017*). Perfectionistic Strivings involves the setting of high personal standards, which has been associated with both positive and negative psychological outcomes (*Limburg et al., 2017*; *Smith et al., 2016*; *Stoeber & Childs, 2010*). The Personal Standards and Organisation subscales of the FMPS, and the Self-Oriented Perfectionism and Other-Oriented Perfectionism subscales of the HMPS consistently load onto the Perfectionistic Strivings factor (*Bieling, Israeli & Antony, 2004*; *Limburg et al., 2017*).

The relationship between perfectionism and burnout has been examined in various non-academic contexts, with Perfectionistic Concerns reliably associated with higher burnout, and aspects of burnout (Exhaustion, Inadequacy, and Cynicism), in many samples, including teachers and junior athletes (e.g., *Hill, 2013*; *Stoeber & Rennert, 2008*). In contrast, Perfectionistic Strivings has been associated with lower levels of burnout,

although the relationship is somewhat inconsistent across the various aspects of burnout (*Hill, 2013*; *Stoeber & Rennert, 2008*).

The few studies that have directly examined the relationship between perfectionism and Academic Burnout have reported broadly consistent findings. Using the Chinese translation of the Frost Multidimensional Perfectionism Scale, Zhang and colleagues (*2007*) demonstrated that Perfectionistic Concerns (measured by Concern over Mistakes, Doubts about Actions, and Parental Expectations subscales) and Perfectionistic Strivings (measured by Personal Standards and Organisation subscales) predicted aspects of Academic Burnout in a sample of 482 Chinese university students. Specifically higher levels of Perfectionistic Concerns predicted more Exhaustion and Cynicism, and less engagement with university. Higher levels of Perfectionistic Strivings predicted lower levels of Exhaustion and Cynicism, and more engagement. These results suggest that individuals with higher levels of Perfectionistic Concerns are more likely to experience higher levels of Academic Burnout, whilst individuals with higher levels of Perfectionistic Strivings are likely to experience lower Academic Burnout and higher study efficacy.

Kljajic and colleagues (*2017*) categorised students as either pure socially prescribed perfectionists, pure self-oriented perfectionists, mixed perfectionists (involving high levels of socially prescribed and self-oriented perfectionism), or non-perfectionistic. Socially prescribed perfectionists (related to Perfectionistic Concerns) were more likely to experience Academic Burnout, as measured separately by Exhaustion, Cynicism, and study efficacy, than non-perfectionists or mixed perfectionists. Moreover, self-oriented perfectionists (related to Perfectionistic Strivings) were less likely to experience Academic Burnout than non-perfectionists and mixed perfectionists. Perfectionism also predicts Academic Burnout among high school students (*Shih, 2012*). Despite the emerging evidence, few studies have considered the possible role of variables that may mediate the association between perfectionism and Academic Burnout. Repetitive Negative Thinking is one potential candidate that warrants investigation.

Repetitive Negative Thinking is an unhelpful continual thought process about past and/or future negative situations, leading to negative emotional states (*McLaughlin & Nolen-Hoeksema, 2011*). It appears to be a risk factor for the development of numerous types of psychopathology (e.g., *McEvoy et al., 2013*). Interestingly, Repetitive Negative Thinking mediates the relationship between perfectionism and various psychological difficulties, including depression (*Flett et al., 2011*), post-traumatic stress disorder (*Egan, Hattaway & Kane, 2014*), and psychological distress (*Macedo et al., 2015*; *O'Connor, O'Connor & Marshall, 2007*). Macedo and colleagues (*2015*) found that Repetitive Negative Thinking partially mediated the positive relationship between Perfectionistic Concerns and aspects of psychological distress (anxiety, depression, anger-hostility, fatigue, and vigor). Interestingly, Repetitive Negative Thinking fully mediated the relationship between Perfectionistic Strivings and depression and fatigue, meaning that Perfectionistic Strivings only predicted higher distress through its relationship with Repetitive Negative Thinking. However, it should be noted that Perfectionistic Strivings did not predict the other aspects of distress in this study. Given the associations between Repetitive Negative Thinking and both
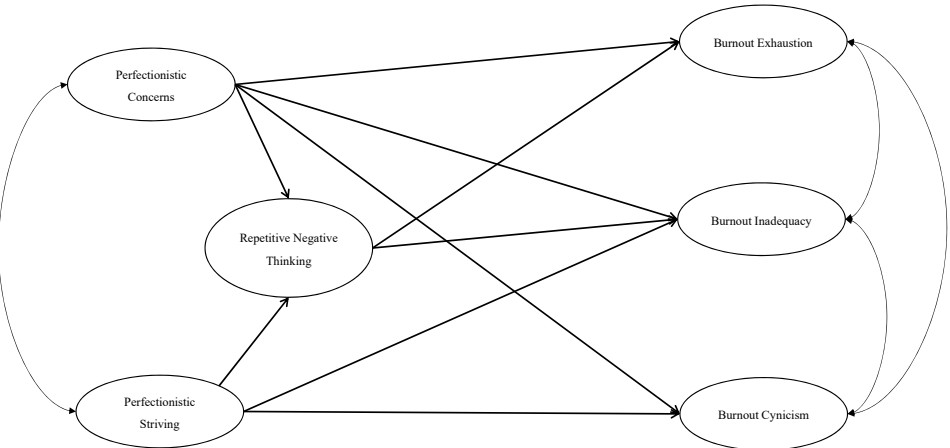

**Figure 1  Hypothesised model.** Complete hypothesised model of direct pathways between perfectionism, burnout, and indirect effects through repetitive negative thinking.

perfectionism and other psychological outcomes, it is plausible that Repetitive Negative Thinking may mediate associations between perfectionism and Academic Burnout.

The current study aimed to (1) replicate associations between perfectionism (Perfectionistic Concerns and Perfectionistic Strivings) and Academic Burnout (Exhaustion, Inadequacy, and Cynicism) and (2) investigate whether Repetitive Negative Thinking mediates the relationships between perfectionism and the elements of Academic Burnout. The hypothesised model is summarised in Fig. 1. The model tested the direct pathways between perfectionism and burnout, as well as the indirect relationships operating via Repetitive Negative Thinking.

## METHOD

### Participants

The sample initially consisted of 215 Australian university students aged 18 and over. However, 89 individuals simply opened and closed the survey without indicating consent or responding to any items, leaving a final sample of 126. The final sample (100 females, 25 males, 1 transgender) ranged in age from 18–69 ($M = 23.64$, $SD = 7.86$), had between 1–8 years experience at university ($M = 3.07$, $SD = 1.56$), and were studying predominantly full-time (104 full time, 22 part time). Of the 126 participants, 57 were recruited from a research participation pool from the School of Psychology at Curtin University and received course credit for participation, while 69 were recruited from other courses via social media, though specific courses were not recorded. There were no significant differences between the mean scores of these two sample groups on any measure used in the study.

### Measures
#### Demographics
Demographic questions measured age, gender, study mode (i.e., full-time or part-time), and number of years completed at university.

*Frost multidimensional perfectionism scale-brief (FMPS-Brief; Burgess, Frost & DiBartolo, 2016).* The FMPS-Brief (*Burgess, Frost & DiBartolo, 2016*) is an eight-item scale measuring two aspects of perfectionism: Perfectionistic Concerns (four items, e.g., *If I fail at university, I am a failure as a person*) and Perfectionistic Strivings (four items, e.g., *I have extremely high goals*). Items are rated on a 5-point Likert scale ranging from *strongly disagree* (1) to *strongly agree* (5). Scores are summed (ranging between 4 and 20), with higher scores indicative of higher levels of Perfectionistic Concerns or Strivings. The Perfectionistic Concerns subscale is strongly, positively correlated with measures of depression, anxiety, hoarding, and worry, indicating good convergent validity (*Burgess, Frost & DiBartolo, 2016; Limburg et al., 2017*). The Perfectionistic Concerns and Strivings subscales have demonstrated good internal consistency in community samples previously ($\alpha = .83$ and $\alpha = .81$, respectively *Burgess, Frost & DiBartolo, 2016*), as well as in the present study ($\alpha = .78$ and $\alpha = .88$, respectively).

*Repetitive negative thinking-10 (RNT-10; McEvoy, Mahoney & Moulds, 2010).* The RNT-10 was adapted from the Repetitive Negative Thinking Scale (*McEvoy, Mahoney & Moulds, 2010*). It contains 10 items assessing engagement with Repetitive Negative Thinking (e.g., *Once you start thinking about the situation, you can't stop*). Items are rated on a 5-point Likert scale ranging from *not true at all* (1) to *very true* (5). Responses from all 10 items are summed (ranging between 10 and 50), with higher scores indicating higher levels of engagement with Repetitive Negative Thinking. The RNT-10 is positively correlated with measures of neuroticism, depression, social anxiety, and worry, indicating convergent validity (*Mahoney, McEvoy & Moulds, 2012*). The RNT-10 demonstrated scale reliability in initial development ($\alpha = .89$, average inter-item correlation $= .44$; *McEvoy, Mahoney & Moulds, 2010*) and subsequent research ($\alpha = .91$, average inter-item correlation $= .49$; *Mahoney, McEvoy & Moulds, 2012*). In the present study the internal consistency of the RNT-10 was high ($\alpha = .94$).

*School burnout inventory (SBI; Salmela-Aro & Näätänen, 2005).* The SBI (*Salmela-Aro & Näätänen, 2005*) is a nine-item questionnaire with three subscales: Exhaustion at School (four items, e.g., *I feel overwhelmed by my schoolwork*), Cynicism Toward Meaning of School (three items, e.g., *I feel that I am losing interest in my schoolwork*), and Sense of Inadequacy at School (two items, e.g., *I often have feelings of inadequacy in my schoolwork*). Items are rated on a 6-point Likert-type scale ranging from *completely disagree* (1) to *strongly agree* (6), with subscale scores calculated by summing the respective items. Scores range between 4 and 24 for the Exhaustion subscale, 3 and 18 for Cynicism subscale, and 2 and 12 for the Inadequacy subscale. Higher scores reflect higher levels of each construct (*Salmela-Aro & Näätänen, 2005*). All items had references to '*school*' changed to '*university*' to reflect the university context in this study. The SBI is correlated with academic achievement, and measures of depression and engagement, demonstrating concurrent validity (*Salmela-Aro et al., 2009*). The SBI demonstrated good internal consistency for subscales ($\alpha = .67$–.80; *Salmela-Aro et al., 2009*). In the present study, the Exhaustion and Cynicism subscales demonstrated good internal consistency ($\alpha = .81$ and .85, respectively), but the internal consistency of the inadequacy subscale was low ($\alpha = .54$).

Table 1  Descriptive statistics and correlation matrix of perfectionism, repetitive negative thinking, and burnout measures (N = 126).

| Measure | Mean (SD) | α | 1 | 2 | 3 | 4 | 5 | 6 | 7 | 8 |
|---|---|---|---|---|---|---|---|---|---|---|
| 1. Perfectionistic concerns | 11.25 (3.61) | .78 | 1 | .48** | .45** | .52** | .23** | .38** | .01 | .16 |
| 2. Perfectionistic strivings | 13.68 (3.76) | .88 | | 1 | .25** | .22** | −.10 | −.11 | .12 | −.01 |
| 3. Repetitive negative thinking | 34.94 (8.89) | .94 | | | 1 | .48** | .22* | .35** | −.11 | .06 |
| 4. Burnout exhaustion | 14.98 (4.60) | .81 | | | | 1 | .40** | .58** | −.01 | .17 |
| 5. Burnout cynicism | 10.56 (4.11) | .85 | | | | | 1 | .65** | −.06 | .11 |
| 6. Burnout inadequacy | 7.54 (2.45) | .54 | | | | | | 1 | −.01 | .12 |
| 7. Age | 23.64 (7.86) | – | | | | | | | 1 | −.01 |
| 8. Gender | – | – | | | | | | | | 1 |

Notes.
*p < .05.
**p < .01.

## Procedure

This study was approved by the Curtin University Human Research Ethics Committee (RDHS-91-16). Participants accessed the anonymous questionnaire online, whereby they viewed a participant information document and provided informed consent before completing the questionnaire, hosted through Qualtrics. Following questionnaire completion, participants were presented with a debriefing document that outlined the purpose of the research and informed them of where to find more information. Participants who required research participation for their course were credited participation points for completing the survey.

## Data analysis

The hypothesised model was tested using path analysis in Mplus. The significance values for both direct and indirect pathways were estimated with a 95% confidence interval using a bootstrapping procedure based on 1,000 draws from the data. Modification Indices (MIs > 20, *Hu & Bentler, 1999*) were examined and theoretically defensible paths were freed. Goodness-of-fit was assessed using the chi-square statistic and degrees of freedom (Chi-square/df), Comparative Fit Index (CFI; values should be ≥0.95), Root Mean Square Error of Approximation (RMSEA; values should be ≤0.06), Tucker-Lewis Index (TLI; values should be ≥0.95), and Standardised Root Mean Square Residual (SRMR; values should be ≤0.08, *Hu & Bentler, 1999*). Although there was no significant correlation between age and gender and perfectionism, Repetitive Negative Thinking, and Academic Burnout, the model was run with and without the control variables of age and gender and the pattern of significant results did not change. Therefore results from the most parsimonious models without control variables are reported. The correlation between Perfectionistic Concerns and Strivings were controlled for, as were the correlations between burnout Exhaustion, Inadequacy, and Cynicism.

## RESULTS

Descriptive statistics and correlations between all variables of interest are summarised in Table 1. Correlations were generally in the expected directions and were small to
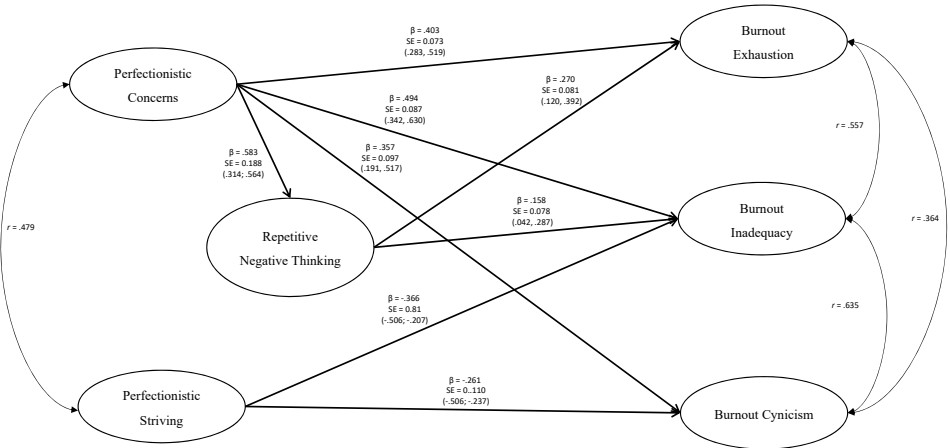

**Figure 2  Final tested model.** Final tested model with only significant pathways between perfectionism, burnout, and repetitive negative thinking considered. Only significant pathways coefficients represented. All coefficients are standardized with 95% confidence intervals in brackets.

moderate in magnitude. The present sample demonstrated scores that were consistent with the non-clinical samples in other studies for both perfectionism scales (*Burgess, Frost & DiBartolo, 2016*), and Repetitive Negative Thinking (*McEvoy, Mahoney & Moulds, 2010*). Scores were slightly higher for the School Burnout Inventory relative to the high school sample reported by *Salmela-Aro & Näätänen (2005)*.

## Path analysis models
### Initial model

A test of the full model indicated a just-identified model, from which fit statistics were not produced (Fig. S1). In this model there were statistically significant direct positive effects of Perfectionistic Concerns on Repetitive Negative Thinking, Burnout Exhaustion, Burnout Inadequacy, and Burnout Cynicism. There was also a statistically significant direct effect of Repetitive Negative Thinking on Burnout Exhaustion and Burnout Inadequacy. Additionally, there were statistically significant direct positive effects of Perfectionistic Strivings on Burnout Inadequacy and Burnout Cynicism. There was no direct relationship between Perfectionistic Strivings and Repetitive Negative Thinking or Burnout Exhaustion. There was also no direct relationship between Repetitive Negative Thinking and Burnout Cynicism.

### Model 2

A second model was run, without the non-significant pathways to test the most parsimonious model. The path analysis revealed good model fit to the data $\chi^2/df = 1.07$, CFI = .999, TLI = .995, RMSEA = .024 (90% CI [.000–.154]), SRMR = .030 (see Fig. 2).

### Direct pathways

There were statistically significant direct positive effects of Perfectionistic Concerns on Repetitive Negative Thinking, Burnout Exhaustion ($p < .001$), Burnout Inadequacy

($p < .001$), and Burnout Cynicism ($p < .001$). There was also a statistically significant direct effect of Repetitive Negative Thinking on Burnout Exhaustion ($p = .001$), and Burnout Inadequacy ($p = .041$). Additionally, there were statistically significant direct positive effects of Perfectionistic Strivings on Burnout Inadequacy ($p < .001$), and Burnout Cynicism ($p = .017$).

### Indirect pathways

There were significant indirect relationships between Perfectionistic Concerns and Burnout Exhaustion ($\beta = .121$, $p = .003$, SE = 0.041, 95% CI [.055–.190]). No other indirect pathways were observed. See Fig. 2 for the final path analysis model with standardised beta, standard error, and 95% confidence intervals for significant pathways.

## DISCUSSION

The aim of the study was to investigate the relationship between perfectionism, Repetitive Negative Thinking, and Academic Burnout. Consistent with previous research (e.g., *Kljajic, Gaudreau & Franche, 2017*), Perfectionistic Concerns was independently associated with all three elements of Academic Burnout among university students. These findings suggest that Perfectionistic Concerns is an important predictor of Academic Burnout among university students and represents a viable target for interventions aimed at reducing burnout in this population. Previous research has demonstrated that cognitive-behavioural programs are effective in reducing problematic Perfectionistic Concerns and that interventions specifically targeting perfectionism can reduce various psychological symptoms, including eating disorder symptoms, generalised anxiety, and depression (e.g., *Rozental et al., 2017*; *Shafran et al., 2017*; also see *Egan, Wade & Shafran, 2011* for a review). Future studies should investigate whether such interventions targeting Perfectionistic Concerns are similarly effective in reducing Academic Burnout in university students. Not only would the results of such studies have considerable practical utility, they would also further understanding of the causal nature of the relationship between Perfectionistic Concerns and Academic Burnout.

Additionally, Repetitive Negative Thinking partially mediated the association between Perfectionistic Concerns and Exhaustion. The indirect relationship between Perfectionistic Concerns and Exhaustion via Repetitive Negative Thinking is broadly consistent with recent research demonstrating that the relationship between perfectionism and various psychological constructs, including burnout, is partially mediated by other variables such as motivation, self-esteem, and coping styles (e.g., *Chang et al., 2015*; *Luo et al., 2016*; *Macedo et al., 2015*). It is also consistent with findings that Repetitive Negative Thinking is a risk factor for multiple psychological symptoms (*McEvoy et al., 2013*) and that it mediates the relationship between perfectionism and various psychological outcomes (e.g., *Flett et al., 2011*).

Furthermore, Repetitive Negative Thinking was also directly associated with burnout Exhaustion and burnout Inadequacy. This highlights the importance of investigating the efficacy of therapeutic programs targeting Repetitive Negative Thinking for students who have high levels of Academic Burnout (particularly that characterised by Exhaustion)

and high levels of Perfectionistic Concerns. Repetitive Negative Thinking can be reduced through targeted interventions (*Watkins, 2018*), or through metacognitive therapy, and this is associated with improvements in several measures of psychological distress (*Johnson et al., 2017*; *McEvoy et al., 2015*). The importance of Repetitive Negative Thinking as a treatment for burnout would critically depend upon whether any modification of a student's Repetitive Negative Thinking is also associated with changes in their experiences of burnout. Future research should investigate the efficacy of such programs in reducing burnout Exhaustion in university students. It is also important to recognise the potential overlap between Repetitive Negative Thinking and depressive disorders (*Nolen-Hoeksema, 2000*) and anxiety disorders (*McEvoy et al., 2013*). Although Repetitive Negative Thinking can be considered a transdiagnostic process (*Harvey et al., 2004*) and is related to burnout, it is unclear whether Repetitive Negative Thinking better explains burnout relative to specific symptoms of depression and anxiety. As such, it would be informative for future research to measure all of these constructs and investigate whether they are independently associated with burnout.

Perfectionistic Strivings was associated with lower levels of Academic Burnout, specifically Inadequacy and Cynicism. This is consistent with previous findings linking constructs closely related to Perfectionistic Strivings with lower burnout (*Kljajic, Gaudreau & Franche, 2017*; *Zhang, Gan & Cham, 2007*). This finding is also consistent with the notion that Perfectionistic Strivings can be a beneficial form of perfectionism (e.g., *Stoeber & Childs, 2010*). However, given evidence that Perfectionistic Strivings might predict negative psychological outcomes in longitudinal studies (*Smith et al., 2016*), this interpretation requires caution. Finally, Perfectionistic Strivings was not indirectly associated with any aspect of Academic Burnout via Repetitive Negative Thinking. Somewhat surprisingly, Perfectionistic Strivings was not associated with Repetitive Negative Thinking. This is inconsistent with the results of Macedo and colleagues (*2015*), who demonstrated that Perfectionistic Strivings predicts depression and fatigue through Repetitive Negative Thinking. Consequently, the relationship between Perfectionistic Strivings and Repetitive Negative Thinking requires further investigation.

Based on the current findings, it is plausible that reducing students' concern regarding mistakes (e.g., Perfectionistic Concerns), may help to reduce Academic Burnout. In many interventions for perfectionism, it is not about reducing an individual's own standards, but rather about promoting a healthy striving for excellence without the individual then basing their self-worth on the striving for or achievement of their high goals (*Egan et al., 2014*; *Handley et al., 2015*). *Nehmy & Wade (2015)* have also tested an intervention for perfectionism within schools that demonstrated improvements in perfectionism, self-criticism, and negative affect at six-months post intervention, with improvements in perfectionism and self-criticism maintained at 12-months post intervention. As Nehmy and Wade demonstrated some preventative effects through their school based intervention, it may also be of use to evaluate whether such a program also provides some preventative effects for academic burnout.

The findings of the present study should be interpreted within the context of the limitations. First, the cross-sectional nature of the data precludes any conclusions regarding the temporal order of the associations. Longitudinal research is clearly needed to address this. Additionally, longitudinal data is required to address the possibility that the relationship between Perfectionistic Concerns, Repetitive Negative Thinking, and Academic Burnout varies depending upon the stage of a student's academic career. It is plausible that the relationship is more pronounced at various time points in semester (e.g., before assignments or exams versus after assessment deadlines) and future research should therefore further examine these factors. Second, the sample was one of convenience and this may limit the generalisability of the findings to the broader student population. Third, the reliability of the Inadequacy subscale was low. Further psychometric assessment of this scale in university samples is needed and findings in the current study related to burnout Inadequacy should be interpreted with caution. Finally, Repetitive Negative Thinking is a theoretically plausible mediator between perfectionism and Academic Burnout because it is a pathway between perfectionism and depression (*Flett et al., 2011*), post-traumatic stress disorder (*Egan, Hattaway & Kane, 2014*), and psychological distress (*Macedo et al., 2015*). However, Repetitive Negative Thinking is only one of many potential factors that might mediate the relationship between perfectionism and Academic Burnout. Future research should examine other potential mediators or moderators. For example, one potential possibility is academic procrastination, which is positively related to Perfectionistic Concerns and negatively related to wellbeing (*Jadidi, Mohammadkhani & Tajrishi, 2011*; *Steel, 2007*). Additionally, individual differences in factors such as imagery, emotion regulation, and coping can exert a strong influence on psychological outcomes (*Holmes et al., 2008*). These variables may be important to consider in the context of Academic Burnout in order to best improve student outcomes.

## CONCLUSIONS

Bearing the limitations in mind, the current study demonstrates that higher levels of Perfectionistic Concerns are associated with greater experiences of Academic Burnout, both directly and indirectly through increased Repetitive Negative Thinking in the case of burnout Exhaustion. Repetitive Negative Thinking was also directly associated with burnout Exhaustion and burnout Inadequacy. In contrast, higher levels of Perfectionistic Strivings are associated with less Academic Burnout. Given that there are treatment programs which can effectively reduce both Perfectionistic Concerns and Repetitive Negative Thinking (*McEvoy et al., 2015*; *Rozental et al., 2017*), future research should focus on understanding the extent to which these programs can alleviate Academic Burnout in university students.

### Funding

The authors received no funding for this work.

## Competing Interests

Mark Boyes is an Academic Editor for PeerJ. The remaining authors declare that they have no competing interests.

## Author Contributions

- David Garratt-Reed and Lana Hayes conceived and designed the experiments, performed the experiments, contributed reagents/materials/analysis tools, authored or reviewed drafts of the paper, approved the final draft.
- Joel Howell conceived and designed the experiments, performed the experiments, analyzed the data, contributed reagents/materials/analysis tools, prepared figures and/or tables, authored or reviewed drafts of the paper, approved the final draft.
- Mark Boyes analyzed the data, contributed reagents/materials/analysis tools, authored or reviewed drafts of the paper, approved the final draft.

## Ethics

The following information was supplied relating to ethical approvals (i.e., approving body and any reference numbers):

Curtin University Human Research Ethics Committee granted ethical approval to carry out the study (Ethics approval number: RDHS-91-16).

## Data Availability

The raw data are provided in a Supplemental File.

## Supplemental Information

Supplemental information for this article can be found online at http://dx.doi.org/10.7717/peerj.5004#supplemental-information.

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
