# Peer review of "Is perfectionism associated with academic burnout through repetitive negative thinking?"

_PeerJ, doi:10.7717/peerj.5004_

## Round 0.1 · original submission · Minor Revisions

· Academic Editor

Minor Revisions

Dear authors

Thank you for you submission. I have now have two reviews and as you note we recommend minor revisions before we can proceed.

Sincerely
Gerhard Andersson

Reviewer 1 ·

Basic reporting

No comment.

Experimental design

No comment.

Validity of the findings

No comment.

Additional comments

The manuscript describes the direct and indirect relationships between the two higher-order dimensions of perfectionism (perfectionistic concerns and perfectionistic strivings) and three elements of academic burnout (exhaustion, inadequacy, and cynicism). Using a measure of repetitive negative thinking, the association between the constructs was also explored by employing path analysis. The results indicate that there is a relation between perfectionistic concerns and academic burnout, and that there was an indirect path through repetitive negative thinking. Overall, the manuscript is well-written, clear, and coherent, and the design and method used seem relevant for the type of research that has been performed. The authors should also be applauded for the fact that they used a non-binary response to gender when recruiting their sample. In sum, the manuscript makes an interesting contribution to the field of perfectionism and academic burnout, however, a few limitations need to be addressed before being considered eligible for publication in the journal PeerJ.

1) The sample used in the study is considered appropriate for its aims, but a major concern is that there might be selections bias due to the large number 89 (41.4%) of missing values. If the authors have any information on those who did not complete all of the measures, this should be reported, such as testing the difference between completers and non-completers. If not, this issue should receive more attention when discussing the limitations.
2) Validated measures of perfectionism, repetitive negative thinking, and academic burnout were used, but it would be useful for the reader to know how the levels on these instruments relate to the population as a whole. Did this particular sample have higher than average levels of, for instance, perfectionism? One concern is that the may be a restriction in range when employing a non-clinical, self-recruited, student sample as the one in the study.
3) Is there any additional information to describe the sample? For example, is the specific university used in the study representative for students in Australia in general? Or did it only include students within a particular discipline, e.g., engineering?
4) At what stage of their education were the students in the sample? One concern is that the relationship between perfectionistic concerns, repetitive negative thinking, and academic burnout could be explained by the level of experience the individuals had in terms of their academic career. It is plausible that novice students are more concerned about making mistakes and thus more prone to academic burnout, while more experienced students have already “learned the ropes” and are less susceptible for negative thinking. The same argument could be used for at what occasion during the semester the study was conducted – could the responses to the measures fluctuate depending on how much the students had to do in their studies?

·

Basic reporting

The reporting is clear and unambiguous. There are minor issues with inconsistency with regard to the use of capital letters and whether the subscales of Perfectionistic Concerns and Strivings are considered as singular or plural.

Although the literature appears adequate, there is a lack of discussion of the role of depression. I am unclear if there is anything specific to Repetitive Negative Thinking or whether the construct of depression could explain the findings equally well. Similarly, it would be helpful to consider the literature on universal school-based programs by Wade and colleagues (Nehmy, T. J., & Wade, T. D. (2015). Reducing the onset of negative affect in adolescents: Evaluation of a perfectionism program in a universal prevention setting. Behaviour research and therapy, 67, 55-63) rather than only the clinical literature since it is more relevant to this study.

The article is professional, raw data are shared, results are relevant to the hypotheses.

Experimental design

This is original primary research within the Aims and Scope of the journal.

The research question is well defined and has the potential to fill an identified knowledge gap. However, there are significant methodological limitations that include:

1) Sample of undergraduate students (many of whom are likely to have been studying psychology). Academic burnout in psychology undergraduates may not generalise to other samples. Information about the representativeness of the sample would be helpful, particularly since the mean age of 23 seems relatively high for undergraduates.
2) No measure of depression despite the very close relationship between repetitive negative thinking and depression
3) Relatively high drop-out; only completer analyses presented. It would be improved with intent-to-treat analysis.

Validity of the findings

The authors themselves highlight concerns with the psychometric properties of the inadequacy subscale of the School Burnout Inventory which calls the reliability and validity of the findings of this subscale into question. The study is sufficiently well described that it would be straightforward to replicate, and the data appear sound. The conclusions are well-stated although there are places (marked on the manuscript) where it is clear the authors are not familiar with the clinical interventions. Suggesting that it might be an idea to try to tackle concern over mistakes and Perfectionistic Concerns rather than Perfectionistic Strivings is not novel, and is a real focus of current therapeutic strategies.

Additional comments

This is a well-written manuscript and study in an important area that has the potential to contribute to the literature. It would benefit from having a clinical expert in the field contribute (such as Sarah Egan who is based at your University) and from more information about the sample to ascertain generalisability; intent-to-treat analyses would be welcome

---

## Round 0.2 · accepted · Accept

· Academic Editor

Accept

Dear authors

I am happy to inform you that your revision addressed the previous comments and you ms now has been accepted for publication.

Sincerely
Gerhard Andersson

#